# The Author’s Contributions to Echocardiography Literature (Part II—1991–2020) [note 1]

**DOI:** 10.3390/children7040034

**Published:** 2020-04-13

**Authors:** P. Syamasundar Rao

**Affiliations:** University of Texas-Houston McGovern Medical School, Children’s Memorial Hermann Hospital, 6410 Fannin Street, UTPB Suite # 425, Houston, TX 77030, USA; P.Syamasundar.Rao@uth.tmc.edu; Tel.: +1-713-500-5738; Fax: +1-713-500-5751

**Keywords:** atrial septal defect, percutaneous closure, buttoned device, total anomalous pulmonary venous connection, balloon pulmonary valvuloplasty, aneurysm of ventricular septum, corrected transposition of the great arteries, patent ductus arteriosus

## Abstract

The author’s contribution up to 1990 was reviewed in part I and the echo contributions from 1991 to 2020 will be reviewed in part II. These include defining the relationship between the quantity of shunt across the atrial septal defect (ASD) and the diameter of ASD by echo and angio on the one side and the stretched diameter of the ASD on the other; echocardiographic assessment of balloon-stretched diameter of secundum ASDs; development of echocardiographic predictors of accomplishment of percutaneous closure of ASDs with the buttoned device, highlighting limitations of echocardiography in comprehensive assessment of mixed type of total anomalous pulmonary venous connection; description of follow-up echocardiographic results of transcatheter closure of ASD with buttoned device; review of ultrasound studies; depiction of collaborative echocardiographic and Doppler studies; echocardiographic appraisal of the outcome of balloon pulmonary valvuloplasty; editorials; ventricular septal aneurysm causing pulmonary outflow tract obstruction in the morphologic left ventricle in corrected transposition of the great arteries; dependability of echocardiographic assessment of angiographic minimal diameter of the ductus; occurrence of supravalvular pulmonary artery stenosis after a Nuss procedure; echocardiographic assessment of neonates who were suspected of having heart disease; role of echocardiographic studies in the appraisal of patent ductus arteriosus in the premature babies; and the role of pressure recovery in explaining differences between simultaneously measured Doppler and cardiac catheterization pressure gradients across outflow tract stenotic lesions.

## 1. Introduction

In the first paper [1], the author’s contributions from 1978 to 1990 were reviewed. In this part, echocardiography contributions from 1991 through 2020 will be reviewed.

## 2. Relationship of the Quantity of Shunt across the Atrial Septal Defect (ASD), Echo Size, and Angiographic Diameter of the ASD with the Stretched Diameter of the ASD

As of early 1990s, device selection for transcatheter closure of the ASDs was mostly on the basis of measured balloon-stretched diameter of the ASD immediately preceding device closure. However, balloon sizing is relatively cumbersome, requiring use of large balloon catheters. Therefore, we examined other methods of evaluation of size of the ASD, such as transthoracic echocardiography, pulmonary-to-systemic flow ratio (Qp:Qs), and angiography to see if the balloon-stretched diameter of the ASD can be predicted [2]. The data of 16 patients with an age range of 7 months and 45 years (median of 4.5 years) who were being assessed for percutaneous closure of ASD were used for this analysis.

The Qp:Qs ratio was 2.6 ± 0.8 with a range of 1.3 to 3.7, the echocardiographic (transthoracic) diameter of the ASD was 9.9 ± 4.1 mm with a range of 4 to 17 mm, and the angiographic size of the ASD was 7.9 ± 2.5 mm (range of 3.8 to 12 mm). The balloon-stretched diameter in these patients was 16.1 ± 5.3 mm with a range of 5 to 23 mm. The two-dimensional (2D) echocardiographic size of the ASD was similar (*p* > 0.1) to the angiographic diameter but was smaller (*p* < 0.01) when compared to the stretched diameter of the ASD. When the relationship of these parameters was look at, the Qp:Qs ratio and ASD diameter by angiography have a noteworthy (*p* < 0.05) correlation with the stretched diameter but with relatively low correlation coefficients: *r* = 0.55 and 0.54, respectively (Figure 1 and Figure 2). The 2D ASD size has an excellent correlation coefficient, *r* = 0.82, *p* < 0.001 with balloon stretched ASD size (Figure 3). Given this relationship, the stretched ASD diameter may be predicted by the following equation: 1.05 × echo + 5.49 mm (Figure 3).

It was concluded that the transthoracic echocardiographic ASD size is useful in the assessment of the stretched ASD size, which in turn may be utilized in the choice of the device size during transcatheter occlusion of the ASD.

## 3. Echocardiographic Assessment of Balloon-Stretched Diameter of Secundum ASDs

As stated in the preceding paper, stretched size of the ASD, measured by balloon sizing during cardiac catheterization, was commonly used in the early 1990s in the selection of the sizes of the devices for percutaneous occlusion of the secundum ASDs. In the preceding study [2], we evaluated the usefulness of Qp:Qs and the angiographic and transthoracic echocardiographic ASD sizes in estimating stretched ASD size and concluded that transthoracic echocardiographic size had the greatest correlation with stretched ASD diameter (*r* = 0.82; *p* < 0.001). The stretched ASD diameter may be calculated by the following formula: (1.05 × echo size in millimeters) + 5.49 mm. In this study [3], we prospectively assessed this formula in the estimation of the stretched ASD size by echocardiographic measurements in a different group of 21 patients with age ranges of 2.5 to 29 years and a median of 4.5 years. 

The echocardiographic diameter of the ASD in these 21 patients was 9.7 ± 3.0 mm, and the stretched ASD size was 15.3 ± 4.0 mm. The predicted stretched ASD size, calculated by the aforementioned formula was 15.7 ± 3.1 mm, which is not appreciably different (*p* > 0.1) from the measured stretched ASD size. The correlation between predicted and measured stretched ASD diameters was good (*r* = 0.9; *p* < 0.001) (Figure 4). The variation between measured and predicted stretched ASD diameter was within 2 mm in all but three patients. 

To improve the accuracy, we suggested that an average of measurements from two subcostal (short and long axis) views be utilized to calculate the ASD size. On the basis of these data, it was concluded that stretched ASD size may be predicted with reasonable accuracy by transthoracic 2D echocardiographic ASD dimensions obtained prior to cardiac catheterization, which in turn may be utilized to select the diameter of device for occlusion of the ASD [3].

While these studies [2,3] were considered useful at that time, subsequent developments, including use of transesophageal (TEE) and intracardiac (ICE) echocardiography and static balloon-sizing may overshadow these findings. Even static balloon-sizing may not be necessary, and TEE or ICE size by using the thick margins of the ASD, eliminating the flail margins, may be used in the selection of the device size for ASD occlusion [4,5].

## 4. Tricuspid Atresia Associated with Persistent Truncus Arteriosus

Echocardiographic finding of tricuspid atresia with persistent truncus arteriosus [6] were reviewed in part I of this review [1] and will not be repeated here.

## 5. Echocardiographic Prediction of Accomplishment of Percutaneous Occlusion of ASD with the Buttoned Device

Percutaneous closure of ostium secundum ASDs with the buttoned device was demonstrated to be safe and effective, as shown in multiple studies published as of the mid-1990s. The decision to insert the device as well as the size of the device selected was mostly on the basis of the balloon-stretched size of the atrial defect, measured during catheterization studies, as alluded to in the two preceding studies [2,3]. However, only subjective measures such as diameter of the ASD and rims of the atrial septum on echo studies have been utilized before cardiac catheterization and balloon sizing. Therefore, we undertook the current investigation to see if objective echocardiographic criteria could be developed that foretell successful closure of the atrial defects [7].

Twenty-nine patients with secundum ASD were assessed for percutaneous closure during a 46-month period ending in August 1992. Successful device implantation was accomplished in 15 patients, and these formed group A while the left over 14 children in whom the device closure could not be performed were assigned to group B. Echocardiograms performed prior to the procedure were analyzed by the authors who were not aware of the results of transcatheter closure attempt. Echocardiographic data acquired were the diameter of the ASD, length of atrial septum (LAS), dimension of the superior and inferior rims in apical and subcostal views (Figure 5), largest jet width by color-Doppler imaging, and the diameters of the right and left atria. Several ratios were derived from these measurements (See Tables I and II from our paper [7].). 

The data so acquired for groups A and B were compared. The LAS, ovality index, diameters of the right and left atria, and superior rims in the subcostal short-axis and four-chamber apical views were comparable (*p* > 0.1) between both groups. The size of the ASD (as measured in all three 2D views and jet width by color-Doppler) was smaller (*p* < 0.05 to < 0.01), the ratio of ASD to LAS was lower (*p* < 0.05), the superior rims as measured in the subcostal long-axis view was shorter (*p* < 0.05), and ratios of superior and inferior septal rims to ASD was larger (*p* < 0.05) in group A than in group B. Multivariate logistic regression analysis and contingency tables showed that diameter of ASD ≤ 15 mm, ASD/LAS ratio ≤ 0.35, and ratio of superior rim to ASD ≥ 0.75 are predictors of good outcome. The greater the number of predictive factors, the higher the probability for successful device occlusion of ASD (See Table VII from our paper [7].). It was concluded that predictive factors for successful occlusion of ASD identified in this investigation are useful in the selection of patients for percutaneous closure of ASD [7].

## 6. Mixed Type of Total Anomalous Pulmonary Venous Return

Total anomalous pulmonary venous connection (TAPVC) is an uncommon congenital heart defect (CHD) constituting approximately 1% of all CHDs. On the basis of site of drainage of the pulmonary veins, the TAPVC patients are classified into supra-cardiac, cardiac, infra-cardiac, and mixed types. The mixed type of TAPVC comprises 5% to 10% of all TAPVCs. We presented echocardiographic and angiographic findings of two cases of mixed TAPVC, with the intent of highlighting the limitations of echocardiography in comprehensive assessment of this rare anomaly [8]. Following echocardiographic and angiographic studies, both children underwent successful surgical correction of their respective defects.

Transthoracic echocardiogram in case 1 showed dilated right atrium (RA) and right ventricle (RV) with a common pulmonary venous confluence (Figure 6a) and a vertical vein emptying into the innominate vein and superior vena cava (SVC) (Figure 6b,c). Dilated coronary sinus (CS) with mosaic color-Doppler flow pattern of pulmonary venous entry was also seen (Figure 6d,e). These findings indicated mixed type of total anomalous pulmonary venous connection. The findings in case 2 were very similar to those seen in case 1. However, not all pulmonary veins were identified and their course and connections to the pulmonary venous confluence could not be established. Therefore, catheterization and cineangiography were undertaken to validate the diagnosis prior to corrective cardiac surgery. Levo-angiographic frames following right pulmonary artery cineangiogram demonstrated entry of right pulmonary veins into the coronary sinus in both cases (Figure 7a,b,d). Direct injection into the left pulmonary vein via a catheter positioned into it via the innominate and vertical veins in the first case (Figure 7c) and on levo-angiogram following left pulmonary artery cineangiogram in the second case (Figure 7e) clearly demonstrated left pulmonary venous drainage via the vertical vein into the systemic venous circuit. Transesophageal echocardiography during surgery in both cases could not delineate the number of pulmonary veins and their connection with the confluence, although we had only access to single-plane TEE probe at that time. 

On the basis of the experience in these two cases, we recommended angiographic definition of all pulmonary veins in mixed type of TAPVC at that time [8]. With today’s availability of better echocardiography machines (compared with mid-1990s) and multi-plane transesophageal probes, angiography may not be needed in most patients suspected of having mixed type of TAPVC. Certainly, today’s cross-sectional imaging can define the anatomy in most patients with mixed TAPVC and angiography may not be necessary.

## 7. Echocardiographic Evaluation of the Results of Transcatheter Closure of the ASD with Buttoned Device

Immediate and short-term results demonstrated that transcatheter closure of the ASD with the buttoned device is feasible, safe, and effective. No long-term results were documented as of early 1990s. Therefore, we evaluated long-term results of buttoned device implantation in 20 of 22 consecutive patients (91%) through August 1992 [9]. The patients were between seven months and 51 years of age with a weight range of 3.6 and 105 kg. The study subjects were grouped into three types on the basis on the type of shunt through the ASD: group I with left-to-right shunt (*n* = 14), group II with presumed paradoxical embolism (*n* = 5), and group III with right-to-left shunt (*n* = 1). Clinical, chest roentgenogram, and echocardiographic evaluation was undertaken at two weeks, at three, six, and twelve months following device occlusion, and subsequently every year. Data at follow-up in these 20 patients were available for review at 29 ± 11 months (range 16 to 52 months) after device implantation. The majority of patients received aspirin (5 to 10 mg/kg/day) orally for 12 weeks following the device placement.

In group I (14 subjects with left-to-right shunt), the Qp:Qs fell from 2.1 ± 0.5 to 1.08 ± 0.2 (*p* < 0.01) following ASD closure. The RV dimension decreased from 2.3 ± 0.6 to 1.7 ± 0.3 cm after occlusion (*p* < 0.01) and stay diminished (1.6 ± 0.46 cm) at latest follow-up evaluation (Figure 8, left panel). The left ventricular (LV) end-diastolic dimensions did not change (Figure 8, right panel). Paradoxical or flat inter-ventricular septal motion was seen in 11 of 14 patients (79%) prior to ASD occlusion which returned to normal at follow-up (Figure 9). Device position was stable (Figure 10) in all patients. 

Small to trivial left-to-right shunts across the atrial septum were seen by color-Doppler in six of 14 patients (43%) the day after ASD occlusion, and trivial shunts persisted in three of fourteen (21%) patients at follow-up. Two of these children had cardiac catheterization one year following device implantation, and no left-to-right shunt was found by oximetry and no shunt was detected on levo-angiographic phase of the pulmonary artery cineangiogram. Generally, small shunts became trivial and trivial shunts disappeared during follow-up. None of the patients had clinical signs suggestive of ASD, and no patient needed surgical closure during this period. 

In group II (5 adult patients, aged 36 to 51 years, with presumed paradoxical embolism), on repeat transesophageal contrast echo study with Valsalva, minimal right-to-left shunt was seen initially (3 months) in two of five patients (40%), which completely disappeared at further follow-up (Figure 11). No patient developed cerebrovascular accident (CVA) or transient ischemic attack (TIA) during a 18-to-43-month follow-up.

In group III, the single patient with CVA due to right-to-left shunt via the residual atrial defect following prior tetralogy Fallot repair had a TIA four months following ASD occlusion. This patient had surgical repair of the residual ASD as per the desire of the patient’s cardiologist. 

There were no fractures of the wire components of device on chest x-rays in any patient. None of the patients in any of the three groups developed bacterial endocarditis.

Based on these data, the conclusion was that effective ASD closure with the buttoned device can be achieved in patients with left-to-right shunt and presumed paradoxical embolism. Modification of the device so as to place the square-shaped patch on the right atrial side may be required to avert cerebrovascular accidents and transient ischemic attacks in subjects with right-to-left shunts [9].

## 8. Ultrasound Reviews 

In collaboration with Dr. Gautam Singh of St. Louis University, several ultrasound reviews were prepared and published [10,11,12]. These include the strengths and pitfalls of 2D and Doppler in the evaluation of left heart outflow obstructions [10], the role of echo-Doppler in interventional pediatric cardiology [11], and assessment of aortic coarctation in adults by echocardiography [12]. The interested reader may review the respective publications [10,11,12].

## 9. Collaborative Echo-Doppler Studies

In collaborative research studies with Dr. Saadeh Jureidini of St. Louis University, multiple echo-Doppler studies were performed, and these include value of innominate vein morphology in the detection of persistent left superior vena cava [13], echo-Doppler in the assessment of flow patterns in the coronary arteries in normal children [14], and a reliable echocardiographic screening method for aberrant coronary arteries [15].

The first of these studies concluded that absent or small left innominate vein (less than 47% of the size of the innominate artery) predicts the existence of a persistent left superior vena cava [13]. The second study demonstrated that coronary artery flow velocity patterns can be recorded and quantified in the majority of children and that the level of left coronary artery flow velocity increases in proportion to the degree of left ventricular hypertrophy [14]. The third study demonstrated that presence of a coronary arterial cross-sectional image in the anterior wall of the aorta in the long-axis view of the LV/aorta (Figure 12) is a useful screening tool to detect aberrant coronary artery in children [15]. Additional details of these studies can be found in the respective publications [13,14,15] for the interested reader.

## 10. Echocardiographic Assessment of the Outcome of Balloon Pulmonary Valvuloplasty

During the evaluation of outcomes of balloon pulmonary valvuloplasty (BPV) for valvar pulmonary stenosis (PS) in 80 patients, three-to-ten-years following the procedure, echo-Doppler studies were utilized [16]. These studies demonstrated a decrease in the Doppler flow velocities and Doppler-derived pressure gradients across the pulmonary valve (Figure 13) and end-diastolic dimensions of the right ventricle (Figure 14). The prevalence and magnitude of pulmonary insufficiency (Figure 15) increased [16]. However, significant volume-overloading of the RV did not occur (Figure 16). 

Based on these data, it was concluded that the long-term results of BPV are excellent and that BPV is the management of option in the treatment of valvar PS; however, our study raises concern regarding development of PI at long-term follow-up [16].

## 11. Editorials

A number of editorials addressing echo-Doppler related subjects were contributed by the author [17,18,19,20,21]; most of these are prepared at the request of the respective chief editors. These editorials addressed cardiac function in juvenile rheumatoid arthritis [17], heart function following transcatheter occlusion of patent foramen ovale [18], LV function after percutaneous occlusion of ASDs [19], atrial electromechanical delay assessed by tissue Doppler imaging in subjects with secundum ASDs [20], and whether intracardiac echocardiography is necessary for monitoring implantation of stents across the site of coarctation of the aorta [21]. For additional details, the interested reader may consult these publications [17,18,19,20,21].

## 12. Ventricular Septal Aneurysm Causing Obstruction of the Pulmonary Outflow Tract in Congenitally Corrected Transposition of the Great Arteries

Congenitally corrected transposition of the great arteries (CCTGA) is a rare CHD. The anatomic, physiological, and clinical aspects of CCTGA with particular attention to sub-pulmonary obstruction of the morphologic LV caused by of the membranous ventricular septal aneurysm in patients with both levocardia and dextrocardia [22,23,24] were described in the past. The ventricular morphology and the sub-pulmonary aneurysm were illustrated angiographically elsewhere [22,23,24], and the echo-Doppler features of these anomalies will be demonstrated in Figure 17, Figure 18, Figure 19 and Figure 20.

## 13. Does Echocardiography Reliably Estimate the Angiographically Measured Minimal Ductal Diameter?

### 13.1. Background and Objectives

With the introduction of trans-catheter occlusion of patent ductus arteriosus (PDA), measurement of the size of the ductus has become important in the choice of the device size for implantation; angiographic minimal ductal diameter is used in the device size selection [25,26,27]. Nonetheless, the selection of subjects for percutaneous closure of PDA is mostly on the basis of echo-Doppler study results during the clinical assessment. The aim of this investigation was to scrutinize the dependability of echo-Doppler studies in estimating the minimal ductal diameter on angiography [28,29].

### 13.2. Subjects and Methods

Forty-seven patients were taken to the cardiac catheterization laboratory with the goal to occlude the PDA during a four-year period from July 2002 to June 2006. All the patients had echo-Doppler studies in a conventional manner prior to the catheterization procedure and aortograms were procured during the cardiac catheterization. Ductal or parasternal short axis views were utilized to determine minimal ductal diameters on 2D and color-Doppler. Straight lateral view images were utilized to measure minimal ductal diameter on angiography.

Thirty-one patients underwent Amplatzer Duct Occluder (ADO) occlusion of PDA, while thirteen subjects had closure with Gianturco coils. PDA occlusion could not be performed in three patients because the PDAs were very large in size (N = 2) or pulmonary vascular obstructive disease (N = 1) was present. The ages of the patients varied between 4 and 303 months; the mean was 55.4 months. The weights ranged between 3.5 kg and 77 kg, and the mean was 17.9 kg. 

### 13.3. Results

The minimal ductal diameter on color-Doppler imaging was 4.26 ± 1.57 mm (Mean ± SD) and is somewhat bigger (*p* = 0.007) than that measured on 2D (3.99 ± 1.37 mm). Both these diameters were larger (*p* < 0.01) than minimal ductal diameter (2.28 ± 1.25 mm) measured by angiography. The study subjects were separated into 5 different age groups as well as into Krichenko types; the discrepancy between echo and angiographic diameters continues to exist (*p* < 0.05 to <0.001) for all age subgroups (Table 1) and for all PDA types. Poor correlation was seen on linear regression analysis between 2D and angiographic (R = 0.288) measurements as well as between color-Doppler and angiographic (R = 0.344) minimal ductal diameters. However, there was a good correlation (R = 0.889) between 2D and color-Doppler minimal ductal diameters.

### 13.4. Conclusions

The presented data point out that both 2D and color-Doppler minimal ductal diameters uniformly overestimate minimal ductal diameters measured on angiography. While estimates by echocardiography of minimal ductal diameter along with echocardiographic LV volume overloading are useful in clinical assessment of PDA patients, the angiographic minimal ductal diameter is important for the choice of the diameter of the device used in percutaneous closure of the PDA.

## 14. Unusual Complication of Supravalvular Pulmonary Artery Stenosis after a Nuss Procedure

An eleven-year-old female patient had a Nuss procedure for treatment of pectus excavatum [30]. A cardiac murmur was heard shortly before the planned Nuss bar removal at the age of thirteen years. Echocardiographic evaluation revealed external compression (Figure 21), causing supravalvar pulmonary stenosis [31]. Removal of the Nuss bar was performed which documented improvement on echo-Doppler studies (Figure 22). Usefulness of echo studies in evaluation of such issues was emphasized.

## 15. Echo-Doppler Assessment of Newborn Babies with Suspected Heart Disease

In a book chapter in the book titled *Perinatal Cardiology: A Multidisciplinary Approach*, echocardiographic evaluation of neonates with suspected heart disease was reviewed [32]. Initially, a brief review of principles of echo-Doppler technique was presented. M-mode and two-dimensional echocardiography, pulsed, continuous wave, and color-Doppler studies from standard views are recorded; subcostal views are most helpful in making a diagnosis in the neonate. In babies with important noncardiac causes of cyanosis or respiratory distress such as persistent pulmonary hypertension, neonatal asphyxia, central nervous system disorders, polycythemia, methhemoglobinemia, hypoglycemia, pulmonary hypoplasia, shock and sepsis, maternal drugs and others, the echo-Doppler studies are useful in confirming normalcy of the heart. The principles of estimation of pulmonary artery pressure were discussed. Then, evaluation of LV function was reviewed. Two easily usable and practical methods for the neonate are LV fractional shortening using M-mode echo (Figure 23) and LV area shortening by 2D echo (Figure 24). It is generally thought that the LV area shortening by 2D echo is better in assessing LV function than LV fractional shortening by M-mode echo, particularly in the neonate and young infant.

Usefulness of echocardiogram in evaluation of infant of a diabetic mother (IDM) (Figure 25, Figure 26 and Figure 27), PDA in premature babies (will be reviewed in the next section), side of the aortic arch in tracheo-esophageal fistula babies (Figure 28), heart defects in Down syndrome (Figure 29), and cardiomegaly (Figure 30 and Figure 31) were elucidated. 

Characteristic echocardiographic features of important cardiac defects such as tetralogy of Fallot, transposition of the great arteries, tricuspid atresia, total anomalous pulmonary venous connection, truncus arteriosus, and hypoplastic left heart syndrome were presented. Echocardiographic images of pulmonary atresia with intact ventricular septum, double-outlet right ventricle, double-inlet left (single) ventricle, Ebstein’s anomaly of the tricuspid valve, and interrupted aortic arch were also shown. In babies without cardiorespiratory distress and a cardiac murmur, the source of murmur may be identified, whether it is a small VSD, branch pulmonary artery stenosis, mild aortic or pulmonary stenosis, or a functional murmur. It was concluded that echocardiography along with Doppler are exceedingly valuable in the assessment of the neonate with suspected and known heart disease. 

## 16. The Role of Echocardiographic Investigation in the Evaluation of Patent Ductus Arteriosus in the Preterm Babies

The PDA produces left-to-right shunt, largely in proportion to the minimal ductal diameter. In premature babies, such shunts produce pulmonary and cardiac compromise. Although the clinical features, chest x-ray, and serum brain natriuretic peptide (BNP) levels are helpful in identifying a PDA, hemodynamically significant PDAs are best detected and quantified with the help of echo-Doppler studies [33,34,35]. The echocardiographic finding of PDA in the premature were reviewed in detail elsewhere [33,34,35] and may be summarized as follows. The left atrium (LA), LA:aortic root (Ao) ratio (<1.4:1), and the left ventricle (LV) are expected to be normal in size in small PDAs, and the systolic function of the LV is preserved. In large PDAs, dilatation of the LA and LV and increase in LA:Ao ratio is (>1.6:1) are seen. At first, the function of the LV is normal or hyperdynamic, and with time, the function of the LV may get worse with consequent elevation in LV end-diastolic and LA pressures. These hemodynamic abnormalities may cause further worsening of the respiratory function. In babies with moderate PDAs, the parameters in are in middle with moderately dilated LA (LA:Ao ratio of 1.4 to 1.6) and LV. In the majority of the babies, the systolic function of the LV remains normal.

In small PDAs, the Minimal ductal diameter (MDD) is small and there is a high Doppler velocity across the PDA (Figure 32), while in large PDAs, the MDD is large and there is a low Doppler velocity across them (Figure 33). In moderately sized PDAs, these parameters are somewhere in between. The pulmonary artery (PA) pressures are usually normal in small PDAs, whereas the PA pressures are probably elevated in large PDAs. While the above statements are mostly accurate, the pressures in the PA also depend upon the magnitude of pulmonary parenchyma disease. Furthermore, in very low birth weight infants, the pressures in the PA may not be increased corresponding to the pulmonary parenchyma disease because of underdeveloped pulmonary vasculature in the premature. 

Lastly, in small PDAs, normal diastolic anterograde flow is seen descending aorta (Figure 34), while in large PDAs, either no normal anterograde diastolic flow or retrograde diastolic flow (Figure 35) is seen in the descending aorta. Most of the medium-sized PDAs have normal diastolic anterograde flow in the descending aorta.

Clinicians caring for premature babies may be able to determine the size of the PDA by the evaluation of the diameter of the LA, LA:Ao ratio, the LV dimension, estimated pressures in the pulmonary artery, MDD, magnitude of Doppler flow velocity across the PDA, and flow patterns in the descending aorta (Table 2). A baby with medium-to-large PDA in association with respiratory compromise may be characterized as having hemodynamically significant PDA [33,34,35].

## 17. Role of Pressure Recovery in Explaining Differences between Simultaneously Measured Doppler and Cardiac Catheterization Pressure Gradients across Stenotic Lesions

Despite an excellent relationship between Doppler and catheterization gradients across obstructive lesions of the heart as we observed/reported [36,37,38,39] as well as by other investigators, the author found significant discrepancies between these values when the author was at the University of Wisconsin. To investigate the reasons for such discrepancy, the author encouraged his echocardiography colleague Dr. Allen Wilson to participate in a study comparing simultaneous catheter and echo measurements during cardiac catheterization. Shortly after the initiation of the study, the author moved to the St. Louis University/Cardinal Glennon Children’s Hospital in St. Louis, MO. Here, the author encouraged another echocardiography colleague Dr. Gautam Singh to participate in such as study. The study was started, and data were secured; the data were presented at multiple scientific societies [40,41,42] but neither the author nor Dr. Singh completed the study manuscript at that time for publication. The following is a summary of the study and data:

Doppler-derived pressure gradients are regularly utilized as a substitute for catheter peak-to-peak gradients (PPGs) for recommending intervention in children with aortic stenosis (AS), PS, and coarctation of aorta (CoA). However, the Doppler gradients do not accurately predict the catheter PPGs. Consequently, misclassification of the severity of lesion may occur with frequent improper consideration for transcatheter or surgical intervention. Pressure recovery (PR) appears to account for most of the difference between Doppler-derived and catheter-measured PPGs in in vitro experimental studies and in investigations in adult subjects. However, clinically important PR in congenital AS, PS, and CoA has not been adequately investigated. Doppler-derived and catheter-measured PPGs were secured simultaneously in a prospective manner in eighty-two consecutive patients with a median age of 12.2 months and a median weight of 7.5 kg with AS (*n* = 30), PS (*n* = 24), and CoA (*n* = 28). Relationship between these values before and after correcting for PR was analyzed. PR was calculated on the basis of fluid dynamic-based equation utilizing echo data. The effect of geometry of the lesion on the magnitude of PR was also examined. 

The Doppler peak instantaneous gradient, corrected for pressure recovery, had considerably better relationship with the catheter-measured PPG (*p* < 0.001) than the uncorrected peak instantaneous and mean Doppler PIGs. The pressure recovery-corrected PIGs were able to predict catheter PPGs with high degree of specificity and accuracy for all lesions with 95 percent confidence limits of 36 to 97 percent and 85 to 100 percent, respectively (*p* < 0.05). The pressure recovery accounted for 4 to 42 percent of the overestimated catheter PPG by Doppler PIG. There was direct relationship (*r* = 0.33 to 0.47) to the valve area with an inverse relationship (*r* = −0.22 to −0.34) to downstream vessel diameter although the correlation coefficients were low. These relationships were more significant (*p* < 0.05) in CoA and PS patients than in AS patients. On the basis of these data, we concluded that considerable PR happens in childhood AS, PS, and CoA and this pressure recovery phenomenon accounts for misclassifying the severity of the lesion by the Doppler-derived PIG. The pressure recovery-corrected Doppler PIGs are more reliable in predicting the catheter-measured PPGs. It also appears that the magnitude PR is dependent on the geometry of stenotic lesion in children.

Recently, a complete manuscript was prepared and submitted for publication and the manuscript was published [43]. 

## 18. Echo Descriptions of Congenital Heart Defects

The author has described echo-Doppler features of several CHDs over the years and these include, pulmonary stenosis [32,44], aortic stenosis [32,44], aortic coarctation [32,44], ASD [4,32,45], VSD [32,46], AVSD [32], PDA [32,33,34,35,45], tetralogy of Fallot and its variants [32,47], transposition of the great arteries [32,48], tricuspid atresia [1,32,49], TAPVC [32,50,51], truncus arteriosus [32,52,53], hypoplastic left heart syndrome [32,54,55], double-inlet left ventricle [32,50], double outlet right ventricle [32,50], pulmonary atresia [50], mitral atresia [32,56], and Ebstein’s anomaly of the tricuspid valve [32,57,58]. For the M-mode, 2D, and Doppler images of these defects, the interested reader may review the respective papers [32,44,45,46,47,48,49,50,51,52,53,54,55,56,57,58].

## Figures and Tables

**Figure 1 children-07-00034-f001:**
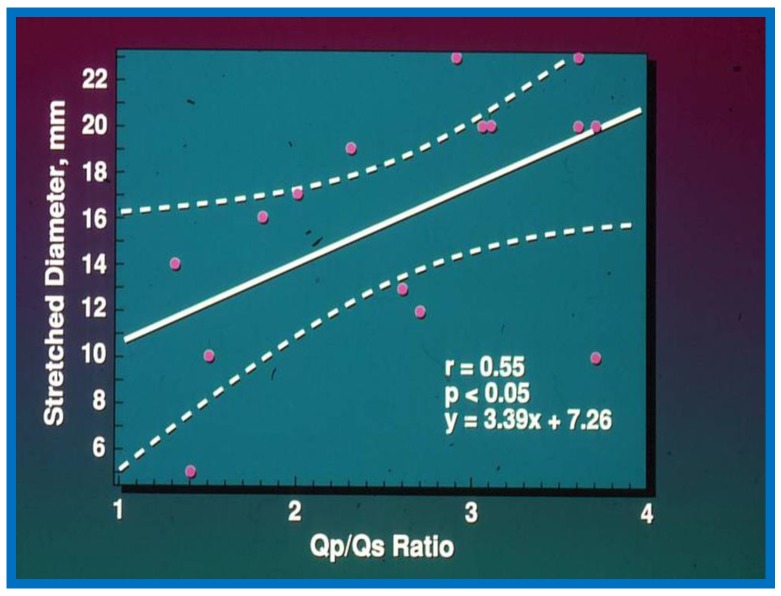
The relationship between pulmonary to systemic flow ratio (Qp/Qs) and stretched diameter of the atrial septal defect is plotted. Regression line (solid line) along with 95% confidence lines (interrupted lines) is shown. Note significant (*p* < 0.05) correlation with an r value of 0.55. Reproduced from Rao P.S., et al. [2].

**Figure 2 children-07-00034-f002:**
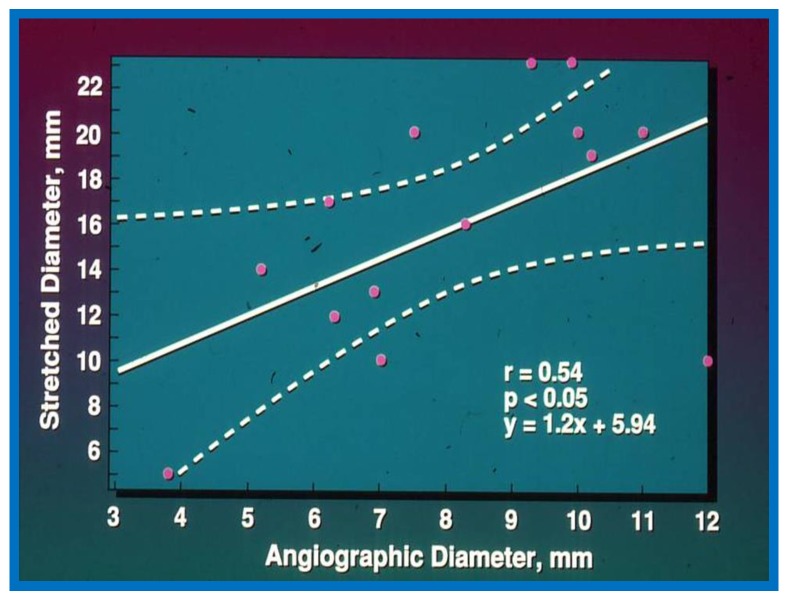
Plot similar to Figure 1 showing the relationship of angiographic diameter of the atrial septal defect with stretched diameter of the atrial septal defect: Note significant (*p* < 0.05) correlation with an r value of 0.55. Reproduced from Rao P.S., et al. [2].

**Figure 3 children-07-00034-f003:**
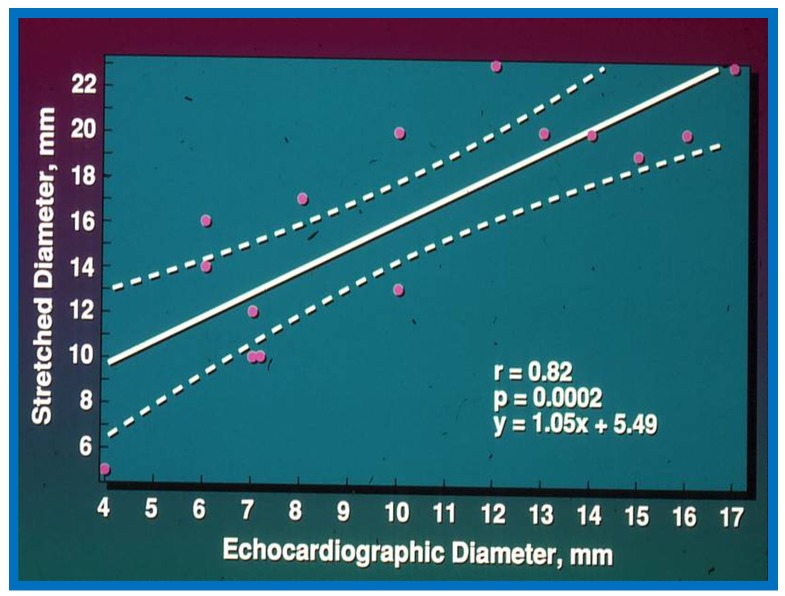
Plot similar to Figure 1 and Figure 2 showing the relationship of transthoracic echocardiographic diameter of the atrial septal defect with stretched diameter of the atrial septal defect: Note significant (*p* < 0.0002) correlation with an r value of 0.82. Reproduced from Rao P.S., et al. [2].

**Figure 4 children-07-00034-f004:**
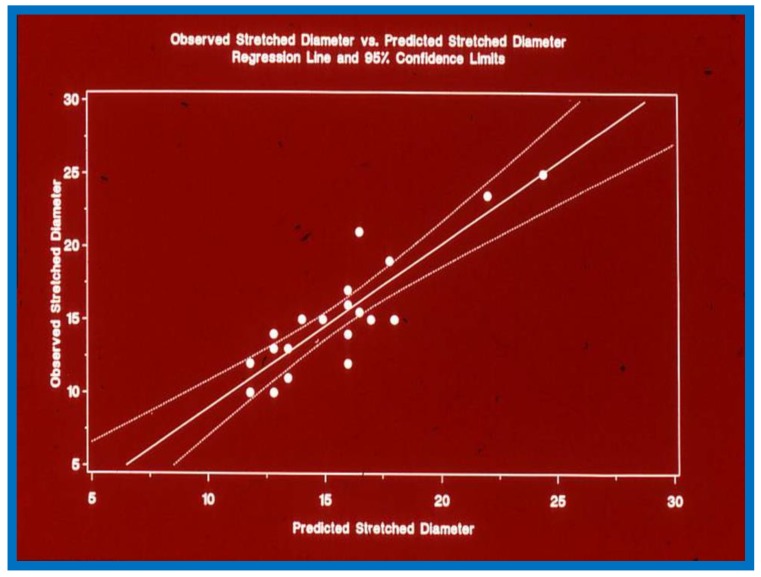
Plot similar to Figure 1, Figure 2 and Figure 3 showing the relationship of predicted atrial septal defect (ASD) diameter by formula (1.05 × echo size in millimeters) + 5.49, with measured stretched size of the atrial septal defect: Note significant (*p* < 0.001) correlation with an r value of 0.9. Reproduced from Rao P.S., et al. [3].

**Figure 5 children-07-00034-f005:**
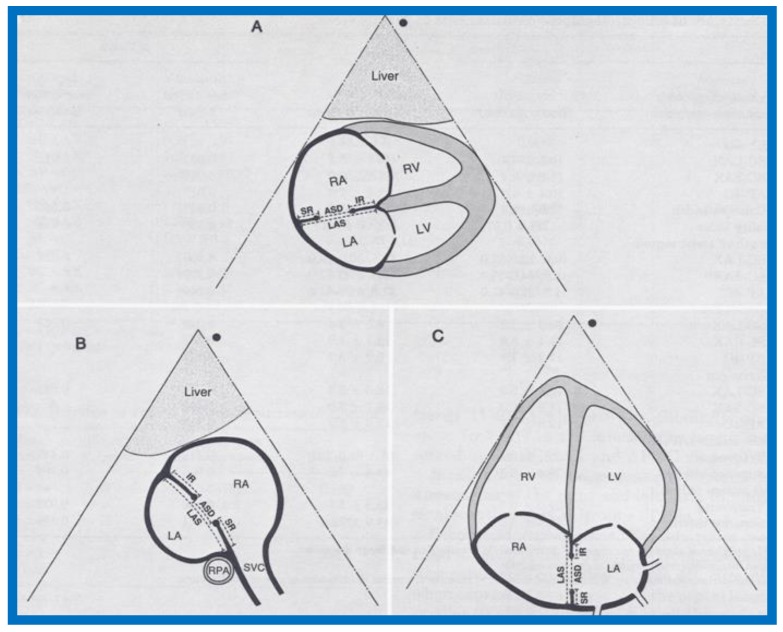
Drawings of 2D echo images of the atrial septum in subcostal long-axis (**A**), subcostal short-axis (**B**), and apical four-chamber (**C**) views demonstrating how measurement of the atrial septal defect (ASD), length of the atrial septum (LAS), superior rim (SR), and inferior rim (IR) are obtained. LA, left atrium; LV, left ventricle; RA, right atrium; RPA, right pulmonary artery; RV, right ventricle; SVC, superior vena cava. Reproduced from Reddy S.C.B., et al. [7].

**Figure 6 children-07-00034-f006:**
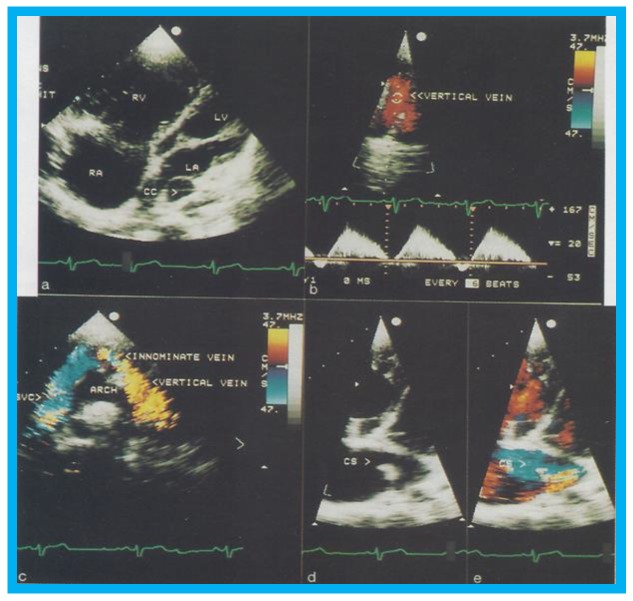
Selected two-dimensional (**a**) and color flow images (**b**–**e**) of a patient (case 1) with mixed type of total anomalous pulmonary venous connection are shown. In (**a**), dilated right atrium (RA) and right ventricle (RV) and common pulmonary venous confluence (CC) are illustrated. In (**b**,**c**), spectral and color flow images demonstrate the vertical vein draining into the innominate vein and superior vena cava (SVC). In (**d**,**e**), dilated coronary sinus (CS) with mosaic color flow pattern of pulmonary venous entry are apparent. ARCH, aortic arch; LA, left atrium; LV, left ventricle. Reproduced from Reddy S.C.B., et al. [8].

**Figure 7 children-07-00034-f007:**
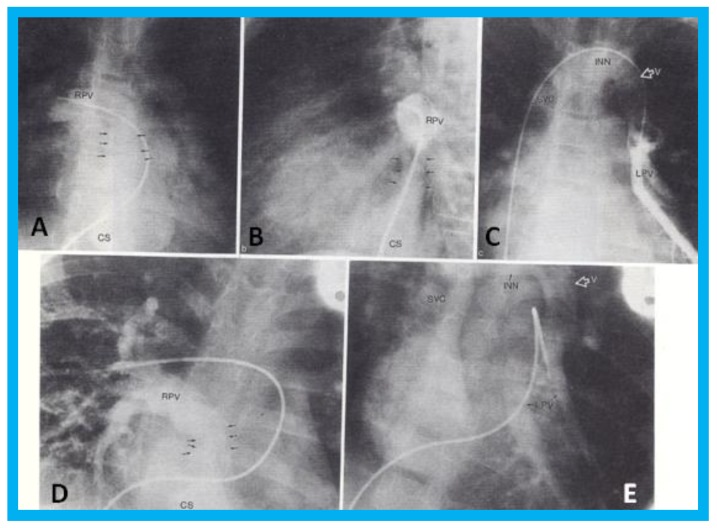
Selected cineangiographic frames from postero-anterior (**A**,**D**,**E**) and lateral (**B**,**C**) views of case 1 (**A**–**C**) and case 2 (**D**,**E**) demonstrating mixed total anomalous pulmonary venous connection. In (**A**,**B**), levo-angiographic frames following right pulmonary artery cineangiogram demonstrated entry of right pulmonary vein (RPV) into the coronary sinus (CS) in case 1. In (**C**) is the selected angiographic frame from left pulmonary vein (LPV) cineangiogram demonstrating pulmonary venous drainage into the vertical vein (V) and then into the innominate vein (INN) and superior vena cava (SVC), also of case 1. In (**D**), levo-angiographic frame following right pulmonary artery cineangiogram demonstrating entry of right pulmonary vein (RPV) into the coronary sinus (CS) in case 2 is shown. In (**E**), levo-angiographic frame following left pulmonary artery cineangiogram demonstrating drainage of left pulmonary veins (LPV) into the vertical vein (V) and then into the INN and SVC of case 2 is shown. In (**A**,**B**,**D**), the connections of the RPVs to the CS are marked with arrows to improve clarity. Reproduced from Reddy S.C.B., et al. [9].

**Figure 8 children-07-00034-f008:**
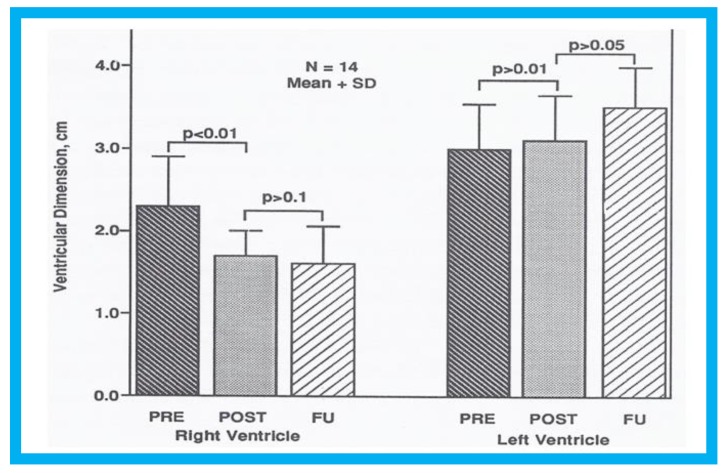
Bar graph illustrating the effect of transcatheter closure of atrial septal defect (ASD) with buttoned device on the left and right ventricular end-diastolic dimensions: Left panel shows that the dimension of the right ventricle fell (*p* < 0.01) immediately following ASD closure. There was an additional but statistically insignificant (*p* > 0.1) decrease in right ventricular dimension at follow-up (FU). The right panel demonstrates that there is no statistically significant change (*p* > 05) either immediately after ASD closure or at FU. POST (on the day following ASD closure), on the day following ASD closure; PRE (prior to ASD closure), prior to ASD closure. Reproduced from Rao P.S., et al. [9].

**Figure 9 children-07-00034-f009:**
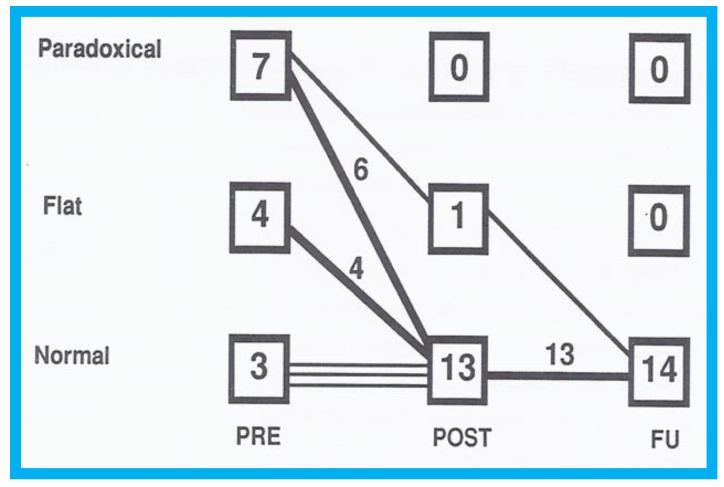
Graph illustrating the effect of transcatheter closure of atrial septal defect (ASD) with buttoned device on the ventricular septal motion: Prior to ASD closure (PRE), the ventricular septal motion is either paradoxical or flat in the majority of patients. Immediately after ASD occlusion (POST), the ventricular septal motion is normal in all but one patient, and at follow-up (FU), the ventricular septal motion returned to normal in all patients. Reproduced from Rao P.S., et al. [9].

**Figure 10 children-07-00034-f010:**
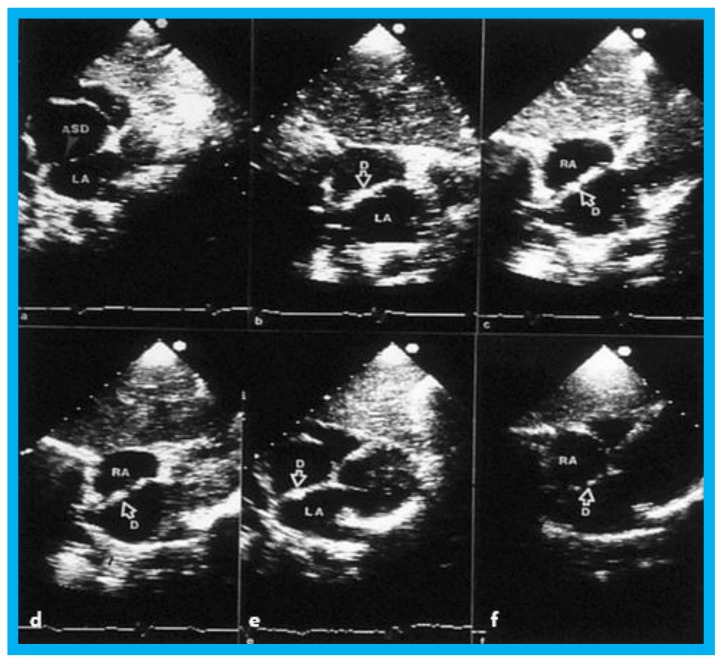
Selected video frames from 2D echo studies in subcostal position of the atrial septum in long-axis view prior to (**a**); immediately after (**b**); and at one (**c**), six (**d**), 12 (**e**), and 24 (**f**) months following atrial septal defect (ASD) closure illustrating the results of transcatheter closure of ASD with buttoned device. Note the position of the device (D) across the ASD (**b**–**f**) during follow-up, the device appears to be incorporated into the atrial septum. On pulsed and color-Doppler studies concurrent with two-dimensional echo studies, there was no evidence for left over shunt (not shown). LA, left atrium; RA, right atrium. Reproduced from Rao P.S., et al. [9].

**Figure 11 children-07-00034-f011:**
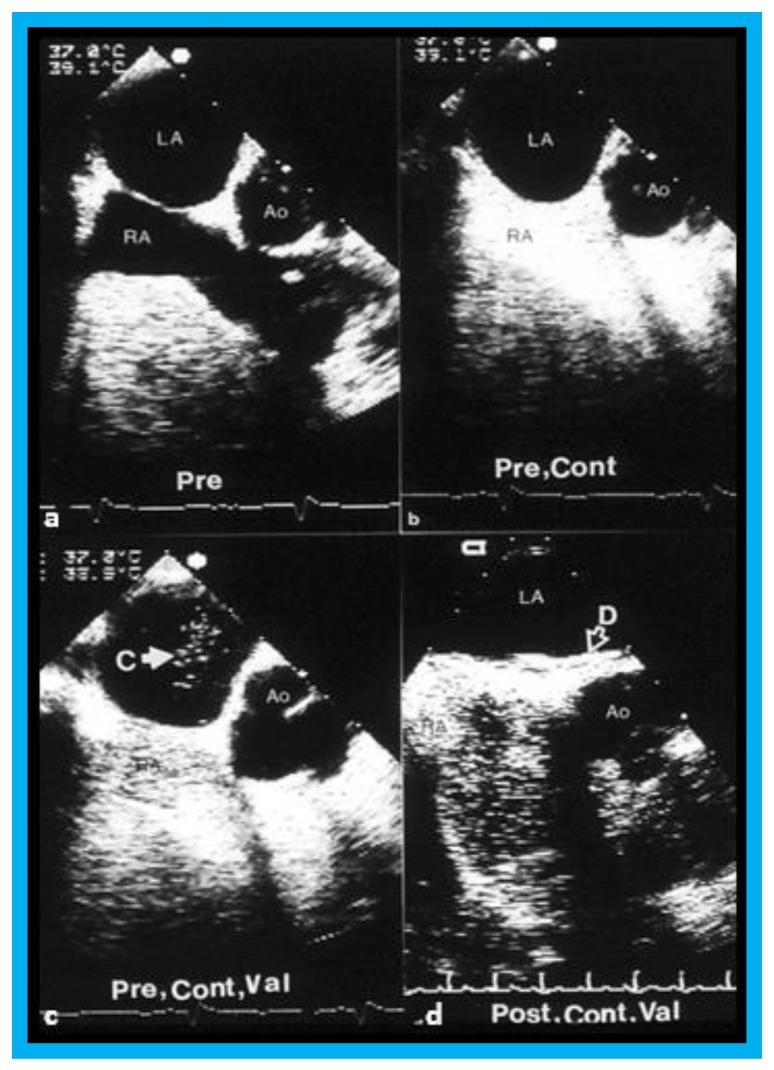
Selected video frames from transesophageal echocardiograms in an adult with presumed paradoxical embolism prior to (Pre) (**a**–**c**) and six months (post) (**d**) after closure of patent foramen ovale with buttoned device: Short axis views illustrate aorta (Ao) in the middle and right atrium (RA) and left atrium (LA) to the right before (**a**) and after (**b**) injection of agitated saline into the RA. Note that there are no contrast bubbles in (**b**). However, with contrast and Valsalva (Val) (**c**), the LA is opacified (C filled arrow). Study similar to c, performed six months following (**d**) (post, Cont, and Val) shows the device (unfilled arrow D) without contrast bubbles in LA. Note that contrast bubbles are seen in RA in (**b**–**d**). Reproduced from Rao P.S., et al. [9].

**Figure 12 children-07-00034-f012:**
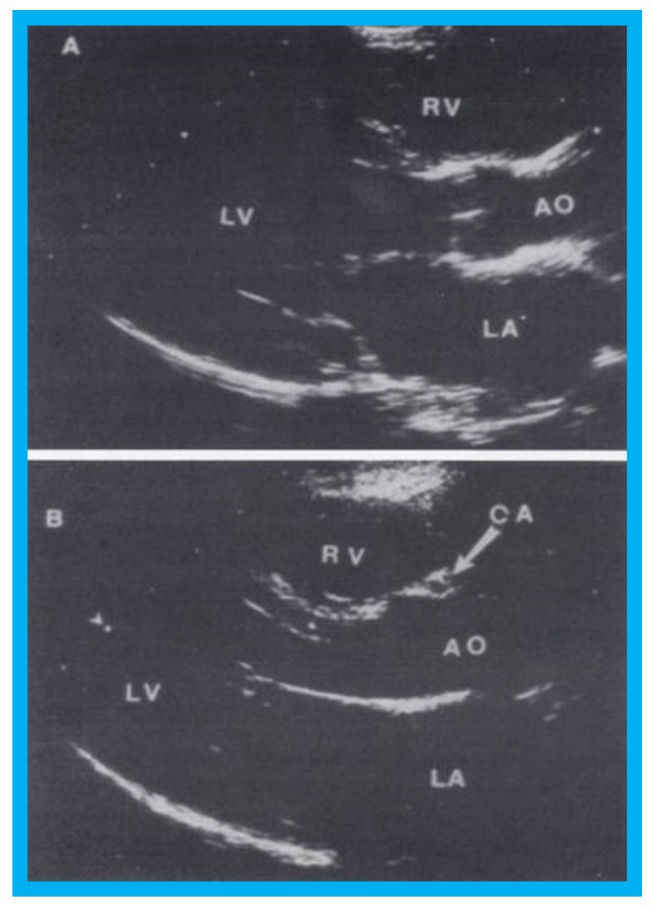
Selected video frames from parasternal long-axis view in a normal child (**A**) and in a child with aberrant coronary artery (**B**): Note a cross-sectional image of the coronary artery (CA) is seen in (**B**) in the anterior wall of the aorta (AO) while such is not seen in (**A**). LA, left atrium; LV, left ventricle; RV, right ventricle. Reproduced from Jureidini S.B., et al. [15].

**Figure 13 children-07-00034-f013:**
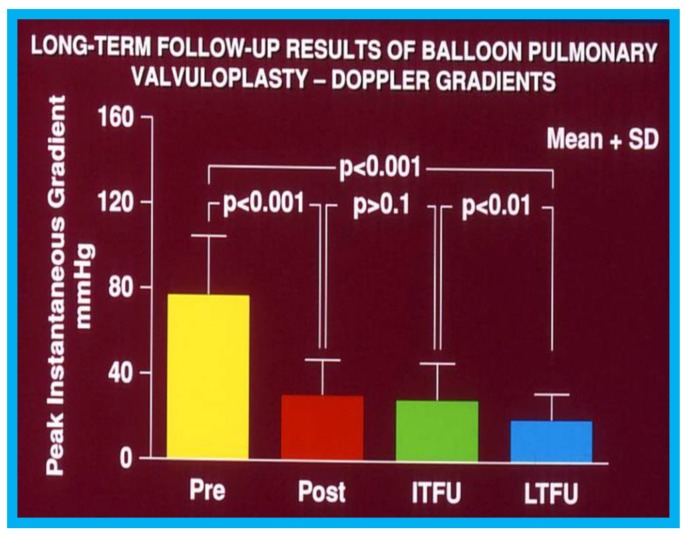
Graph showing calculated Doppler peak instantaneous gradients prior to (Pre) and one day after (Post) balloon pulmonary valvuloplasty (BPV) and at intermediate-term (ITFU) and long-term (LTFU) follow-up: Significant decrease (*p* < 0.001) following BPV (Pre vs. Post) occurred which did not change (*p* > 0.1) at ITFU. But, at LTFU, there was a further decrease (*p* < 0.001) in the Doppler-calculated gradients. Mean + standard deviation (SD) are depicted. Reproduced from Rao P.S., et al. [16].

**Figure 14 children-07-00034-f014:**
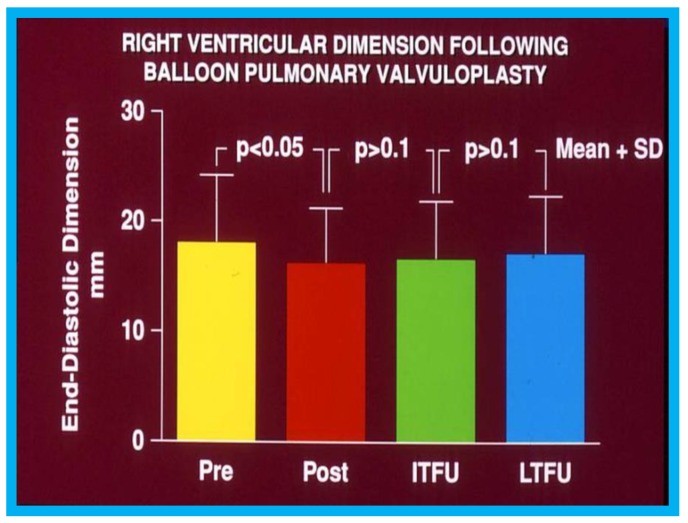
Graph showing end diastolic dimension of the right ventricle (RV) prior to (Pre), one day following (Post), at intermediate-term (ITFU), and at long-term (LTFU) follow-up after balloon pulmonary valvuloplasty (BPV). A significant reduction (*p* < 0.05) in RV size immediately after BPV was seen. No further change at ITFU and LTFU occurred. Increased (*p* < 0.05) prevalence of flat septal motion was observed at LTFU (see Figure 16). None of the patients had paradoxical septal motion. Mean + standard deviation (SD) are depicted. Reproduced from Rao P.S., et al. [16].

**Figure 15 children-07-00034-f015:**
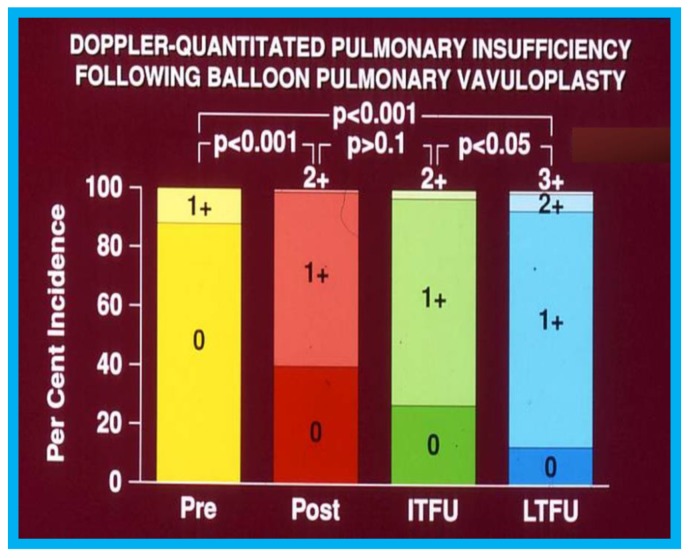
Bar graph showing Doppler-graded pulmonary insufficiency prior to (Pre) and one day after (Post) balloon pulmonary valvuloplasty and at intermediate-term (ITFU) and long-term (LTFU) follow-up: A gradual but significant increase (*p* < 0.05 to *p* < 0.001) in the incidence of pulmonary insufficiency is seen. Reproduced from Rao P.S., et al. [16].

**Figure 16 children-07-00034-f016:**
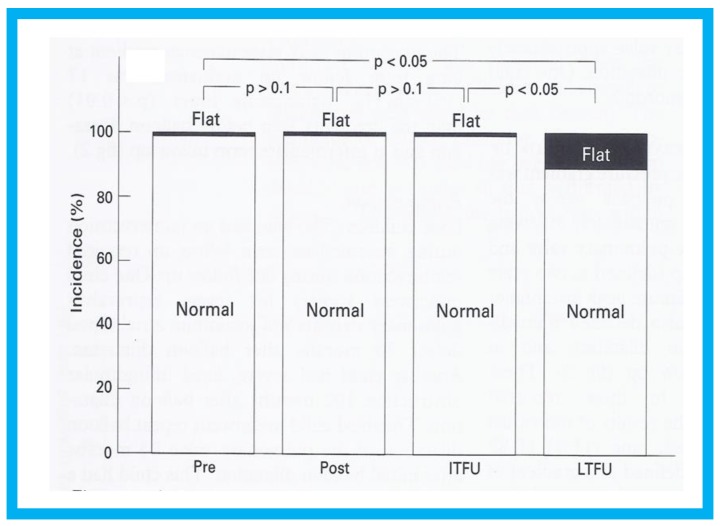
Bar graph showing the prevalence of inter-ventricular septal motion prior to (Pre) and one day after (Post) balloon pulmonary valvuloplasty and at intermediate-term (ITFU) and long-term (LTFU) follow-up: Note significant increase (*p* < 0.05) in the incidence of flat septal motion at LTFU. No patient was observed to have paradoxical septal motion. Reproduced from Rao P.S., et al. [16].

**Figure 17 children-07-00034-f017:**
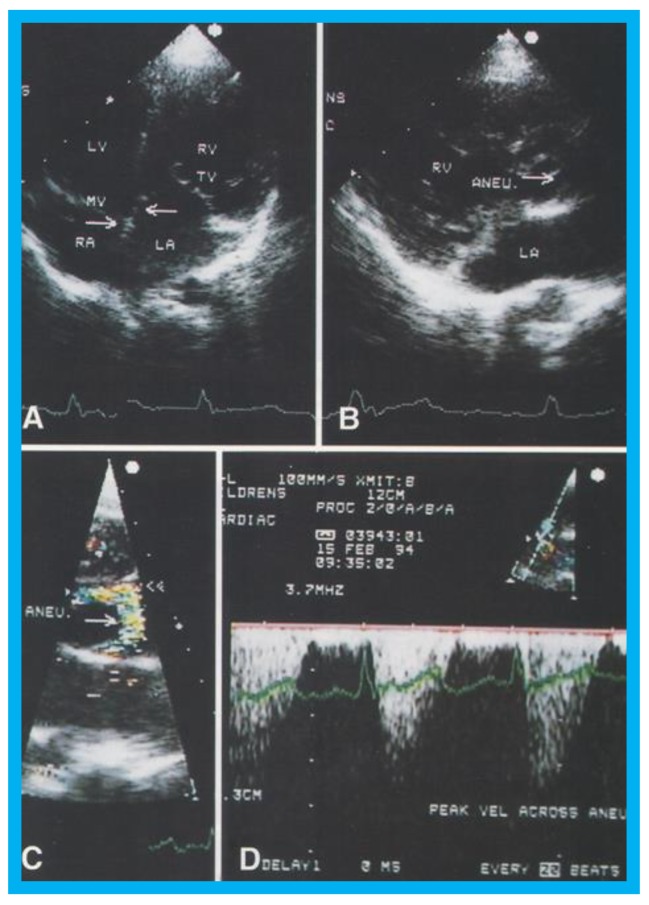
(**A**) Selected video frame from apical four chamber projection indicating reversal of the ventricles: The attachment of the mitral valve (MV) leaflet is higher than that of the tricuspid valve (TV), suggesting that the morphologic right ventricle (RV) is on the left side and that the morphologic left ventricle (LV) is on the right side. The right atrium (RA) drains into morphologic LV, and the left atrium (LA) is connected to the morphologic RV. Also note that the medial leaflet of the TV is plastered on to the ventricular septum suggesting Ebstein’s type of morphologic TV. (**B**,**C**) Selected video frames from parasternal long axis view demonstrate the aneurysm (Aneu) with color-Doppler turbulence (**C**). (**D**) This frame shows a high peak Doppler flow (3.5 m/s), which indicates a peak instantaneous gradient of 49 mmHg. Reproduced from Reddy S.C.B., et al. [22].

**Figure 18 children-07-00034-f018:**
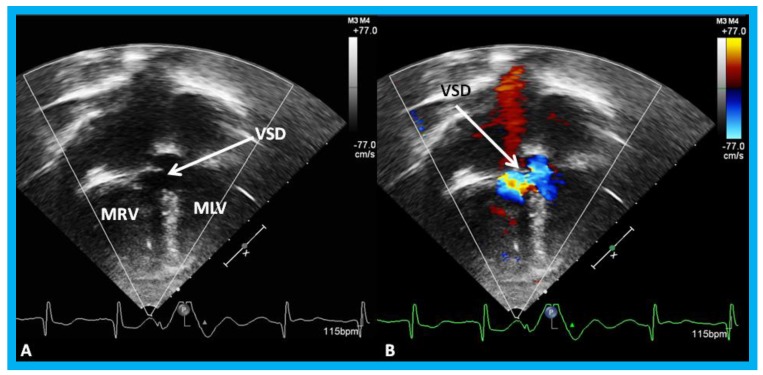
Selected video frames by two-dimensional (**A**) and with color flow imaging (**B**) from apical four-chamber view from the right chest demonstrating morphologic right (MRV) and morphologic left (MLV) ventricles and a large VSD (ventricular septal defect) in a patient with dextrocardia. Reproduced from Yarrabolu T.R., et al. [23].

**Figure 19 children-07-00034-f019:**
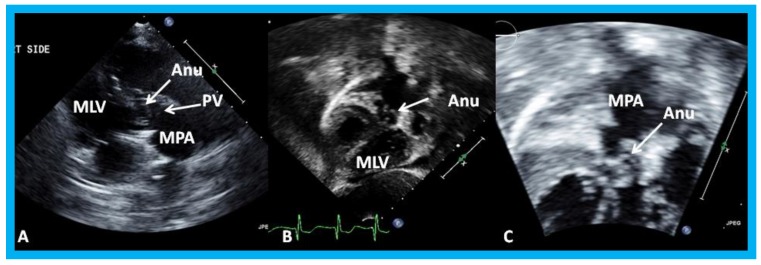
Selected video frames from a right parasternal (**A**) and subcostal four chamber (**B**) views of the morphologic left ventricle (MLV) demonstrating aneurysm (Anu) projecting into the MLV outflow tract, producing obstruction: The Anu is located just below the pulmonary valve (PV). The main pulmonary artery (MPA) is dilated. (**C**) An enlarged view of B illustrating the aneurysm and MPA dilatation. Reproduced from Yarrabolu T.R., et al. [23].

**Figure 20 children-07-00034-f020:**
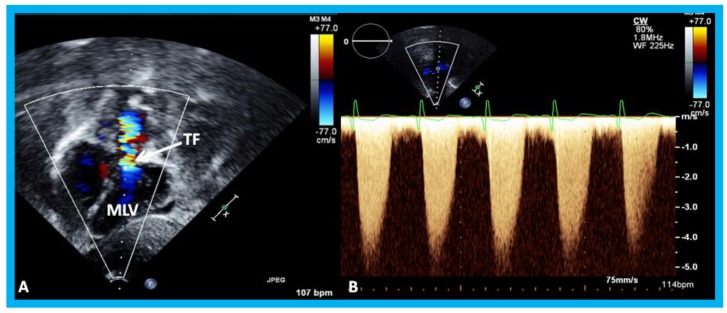
(**A**) Selected video frame from a subcostal four chamber view (similar to Figure 19B) with color-Doppler demonstrating turbulent flow (TF) in the pulmonary outflow tract. (**B**) Continuous wave Doppler recording across the pulmonary outflow tract demonstrates peak Doppler velocity of approximately 5 m/sec suggesting significant obstruction. MLV, morphological left ventricle. Reproduced from Yarrabolu T.R., et al. [23].

**Figure 21 children-07-00034-f021:**
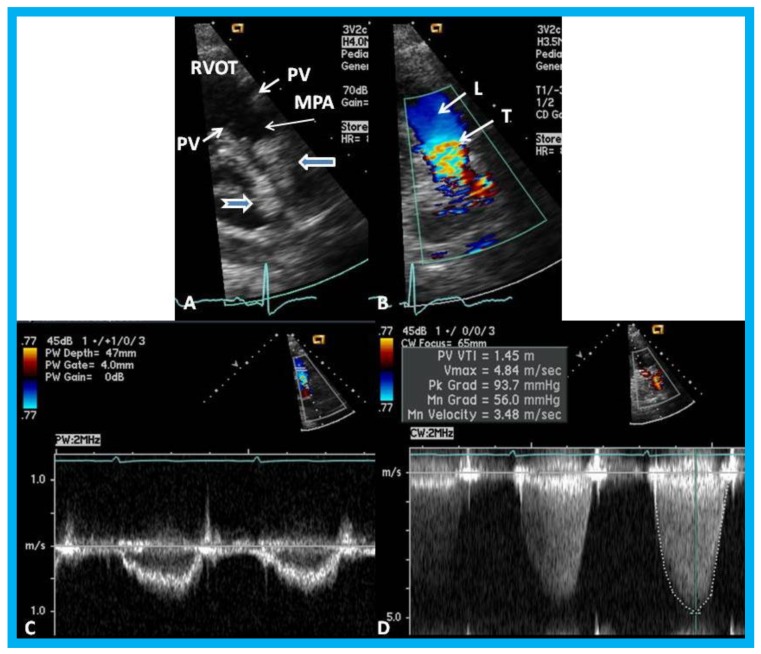
(**A**) Selected video frame from a parasternal short axis view showing echo dense structures (thick blue arrows) within and outside the main pulmonary artery (MPA): Pulmonary valve (PV) leaflets (small arrows) are shown and appear normal. The right ventricular outflow tract (RVOT) and proximal MPA are free of any echo-dense structures. (**B**) Color-Doppler mapping of the same structures as in panel A shows normal laminar (L) flow in the RVOT and proximal MPA and turbulent (T) flow starting in the proximal MPA, indicating obstruction. (**C**) Pulse Doppler sampling from the proximal MPA, which shows normal flow velocity. (**D**) Continuous wave Doppler sampling demonstrating high velocity flow across the MPA with a calculated peak instantaneous gradient of 93.7 mmHg and a mean gradient of 56 mmHg, indicating severe obstruction. Reproduced from Mazur L., et al. [31].

**Figure 22 children-07-00034-f022:**
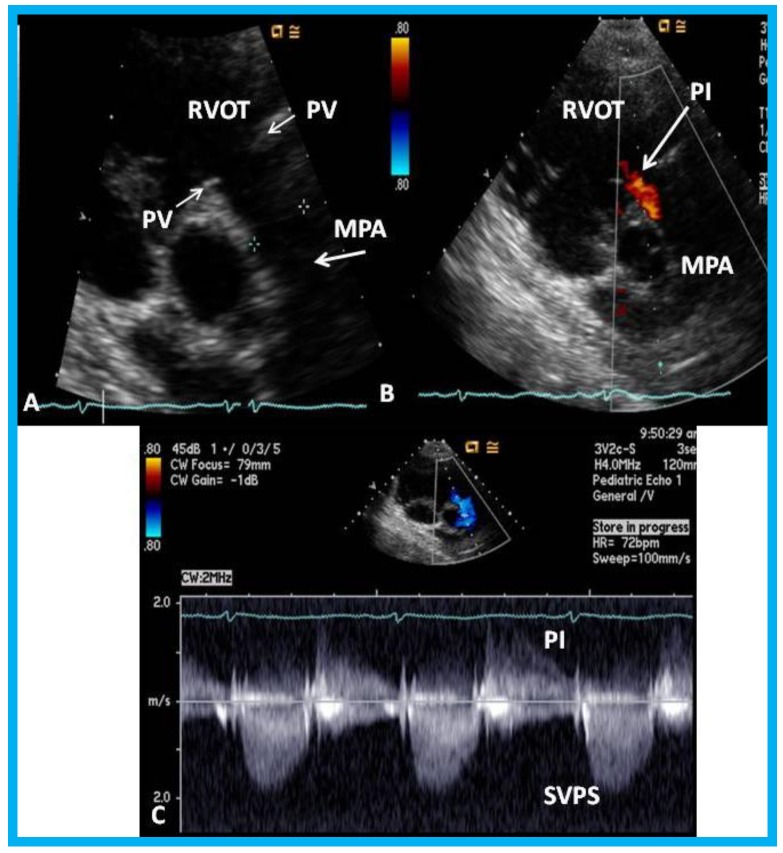
Echo-Doppler studies performed five months after removal of the Nuss bar: (**A**) Selected video frame from a parasternal short axis view demonstrating no echo dense structures in the right ventricular outflow tract (RVOT) and main pulmonary artery (MPA) that was seen prior to removal of the Nuss bar (Figure 20A). Pulmonary valve (PV) leaflets (arrows) are shown. (**B**) Color-Doppler mapping of the same structures as in panel A shows mild pulmonary insufficiency (PI) (arrow). (**C**) Continuous wave Doppler sampling demonstrating low Doppler flow velocity across the MPA with a calculated peak instantaneous gradient of 15 mmHg, indicating minimal supravalvular pulmonary stenosis (SVPS) and mild pulmonary insufficiency (PI). Reproduced from Mazur L., et al. [31].

**Figure 23 children-07-00034-f023:**
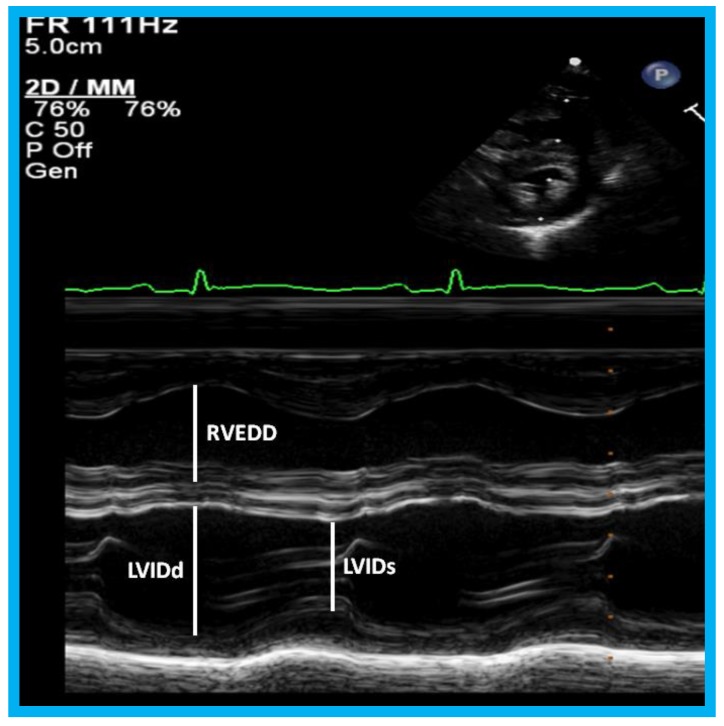
M-mode recording of the ventricles demonstrating measurements of left ventricular internal dimension in end-diastole (LVIDd) and right ventricular end-diastolic dimension (RVEDD), both measured at the onset of the QRS complex of the simultaneously recorded electrocardiogram (ECG): The left ventricular internal dimension in systole (LVIDs) is measured at end-systole. The data are compared with normal values, and z score is determined. These data are also used for calculation left ventricular fractional shortening: FS = {(LVIDd − LVIDs)/LVIDd} 100 (where FS is fractional shortening, LVIDd is left ventricular end-diastolic dimension, and LVIDs is left ventricular end-systolic dimension.) to assess the left ventricular systolic function.

**Figure 24 children-07-00034-f024:**
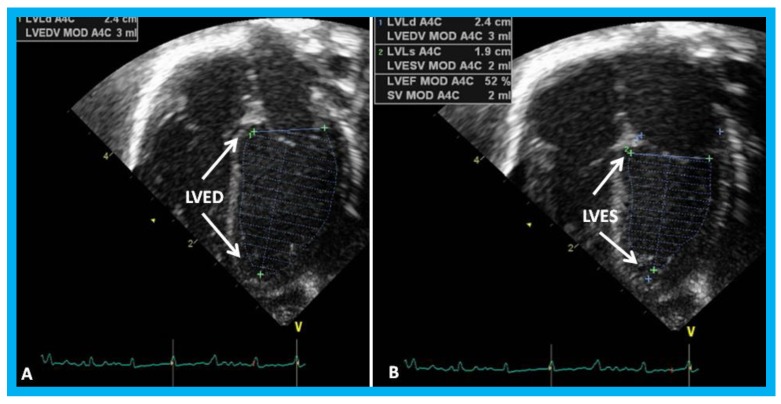
Apical four-chamber views of the left ventricle (LV) in end-diastole (LVED) in (**A**) and end-systole (LVES) in (**B**) demonstrating calculation of area shortening of the LV using Simpson’s rule: AS = (LVAd − LVAs)/LVAd (where AS is area shortening, LVAd is LV area in diastole, and LVAs is LV area in systole). The LV area shortening is 52% (see insert in (**B**)); normal values are 40 to 60%.

**Figure 25 children-07-00034-f025:**
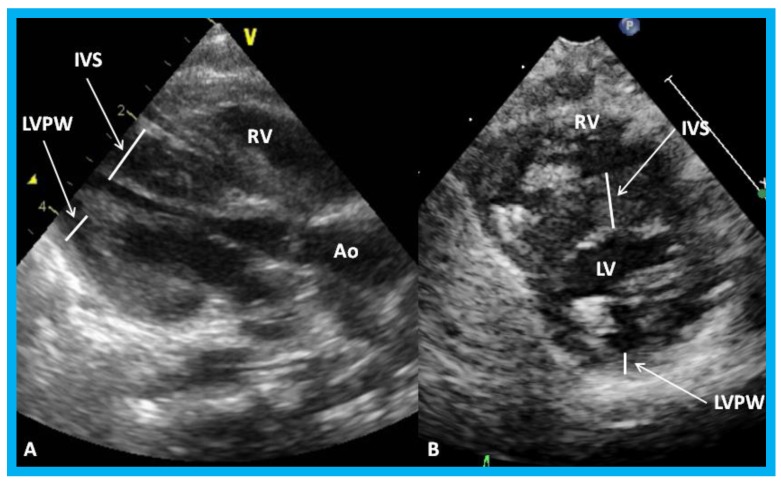
Parasternal long (**A**) and short (**B**) axis views of the left ventricle (LV) demonstrating markedly thickened interventricular septum (IVS) in an infant of a diabetic mother. Ao, aorta; LVPW, LV posterior wall thickness; RV, right ventricle.

**Figure 26 children-07-00034-f026:**
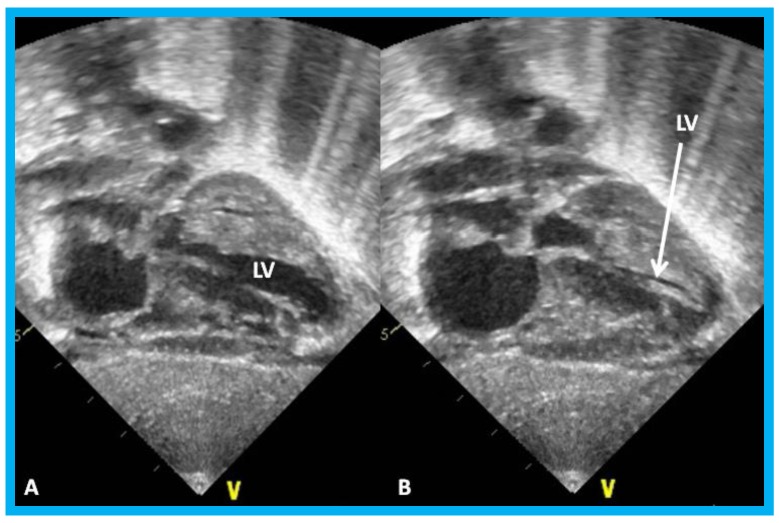
(**A**,**B**). Subcostal views of the left ventricle (LV) in an infant of a diabetic mother showing complete obliteration of the LV cavity in systole (arrow in (**B**)).

**Figure 27 children-07-00034-f027:**
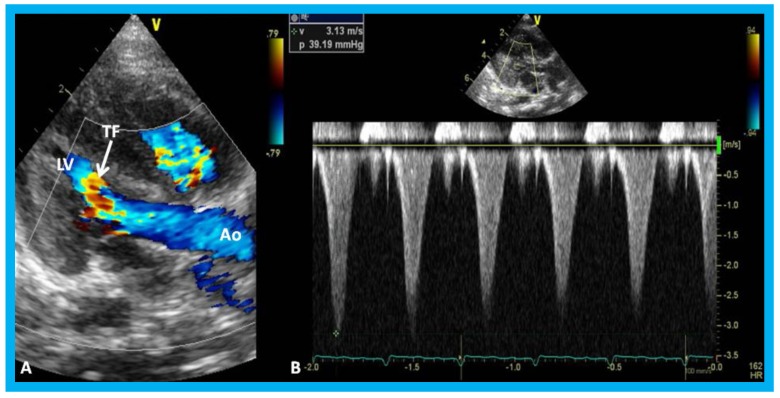
(**A**) Parasternal long axis, two-dimensional, and color-Doppler images demonstrating turbulent flow (TF) in the outflow tract of the left ventricle (LV) in an infant of a diabetic mother. (**B**) Continuous wave Doppler recording demonstrates a peak instantaneous gradient of 39 mmHg (see the insert in (**B**)): Note the triangular pattern of the Doppler recording indicative of subaortic obstruction. Ao, Aorta.

**Figure 28 children-07-00034-f028:**
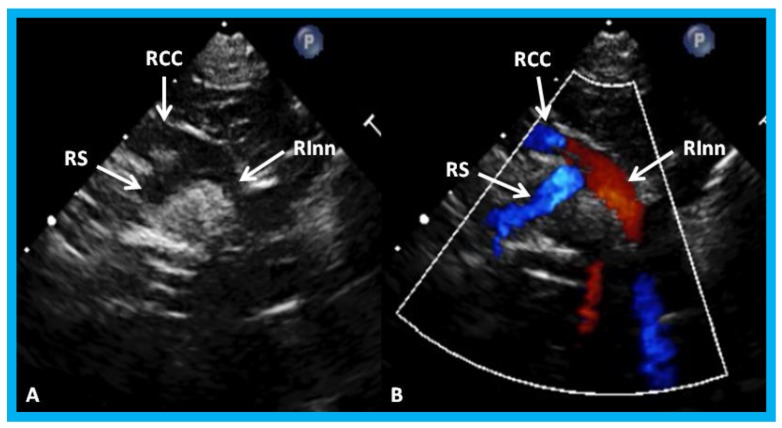
High parasternal views of the right innominate (RInn) artery demonstrating its division into right common carotid (RCC) artery and right subclavian (RS) artery by 2D (**A**) and color flow imaging (**B**): These data suggest left aortic arch.

**Figure 29 children-07-00034-f029:**
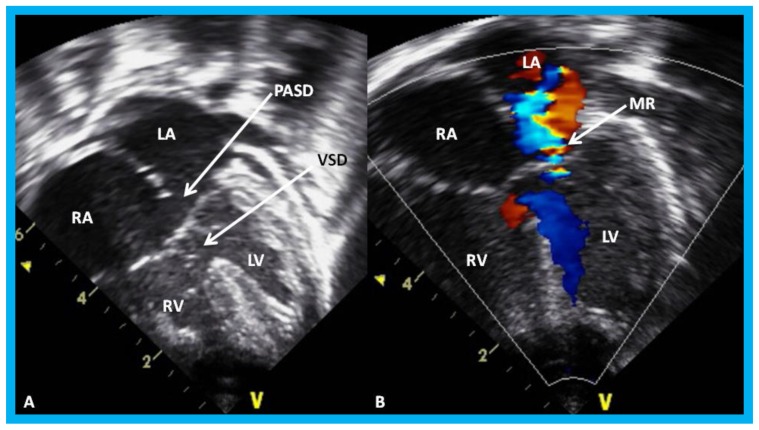
Apical four-chamber view of the heart in baby with Down syndrome demonstrating atrioventricular septal defect in 2D (**A**) and color-Doppler imaging (**B**): Ostium primum atrial septal defect (PASD) and ventricular septal defect (VSD) are shown (arrows) in (**A**) and mitral regurgitation (MR) in (**B**) (arrow). LA, left atrium; LV, left ventricle; RA, right atrium; RV, right ventricle.

**Figure 30 children-07-00034-f030:**
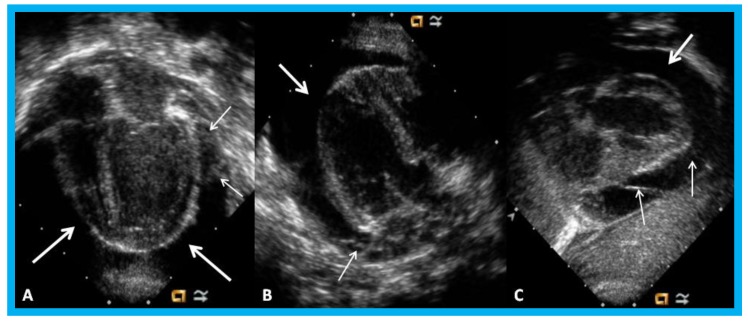
Apical four-chamber (**A**,**B**) and parasternal short axis (**C**) views of the heart in an infant with cardiomegaly showing a significant pericardial effusion (thick arrows): Fibrin strands in the effusion are shown (thin arrows).

**Figure 31 children-07-00034-f031:**
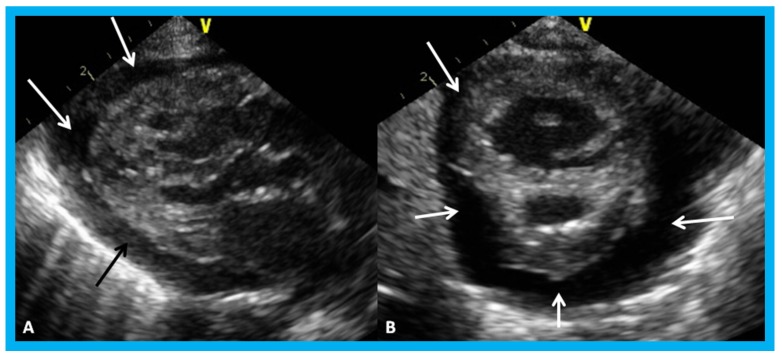
Parasternal long (**A**) and short (**B**) axis views of the heart in a baby with cardiomegaly demonstrating a large pericardial effusion (arrows).

**Figure 32 children-07-00034-f032:**
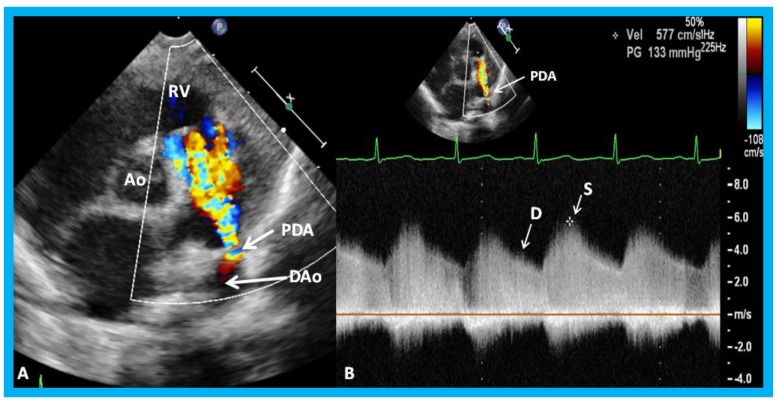
Parasternal short axis echo-Doppler imaging (**A**) demonstrating patent ductus arteriosus (PDA) with left-to-right shunt and high Doppler flow velocity (**B**) across the PDA suggestive of normal/low pulmonary artery pressure. Ao, aorta; D, diastolic; DAo, descending aorta; RV, right ventricle; S, systolic.

**Figure 33 children-07-00034-f033:**
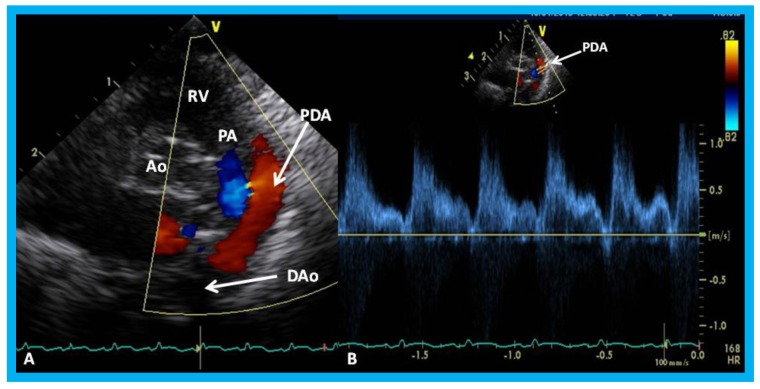
Parasternal short axis echo-Doppler imaging (**A**) demonstrating patent ductus arteriosus (PDA) with left-to-right shunt and low Doppler velocity (**B**) across the PDA suggestive of elevated pulmonary artery pressure. Ao, aorta; DAo, descending aorta; PA, pulmonary artery; RV, right ventricle.

**Figure 34 children-07-00034-f034:**
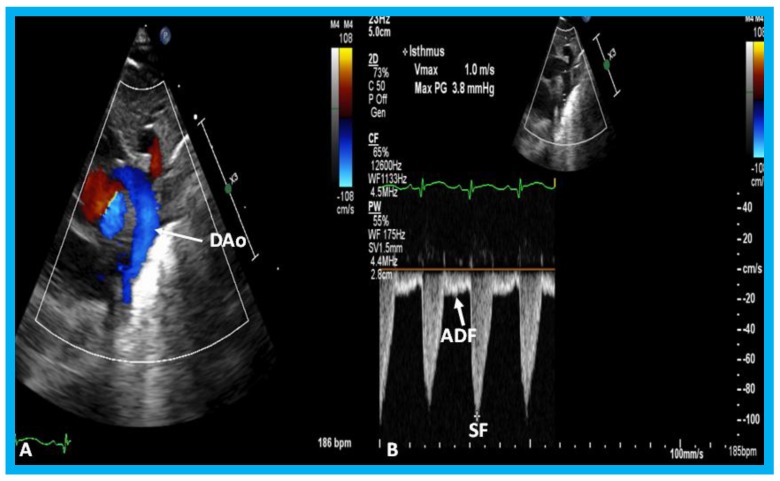
(**A**) Echocardiographic frame from a suprasternal notch view illustrating laminar flow in the descending aorta (DAo) in a premature infant with a small ductus (not shown). (**B**) Continuous wave Doppler recording in the same infant shows normal systolic flow (SF) (*) and normal anterograde diastolic flow (ADF) in the DAo; the diastolic flow is seen below the baseline.

**Figure 35 children-07-00034-f035:**
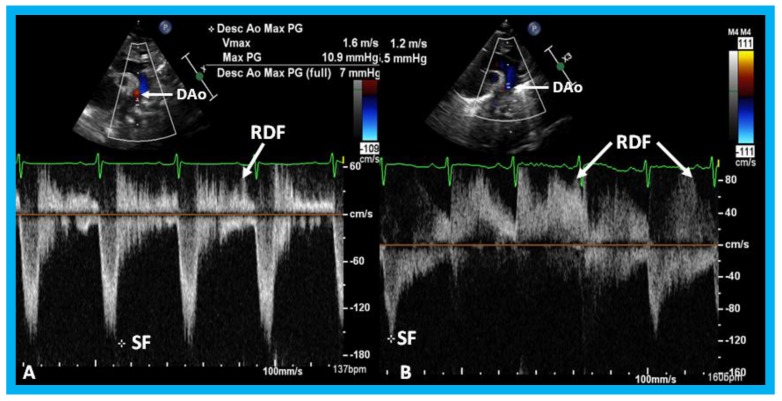
Doppler interrogation from suprasternal notch position in two different premature infants (**A**,**B**) with large patent ductus arteriosus (PDA) show retrograde diastolic flow (RDF) in the descending aorta (DAo), suggesting that these PDAs are likely be hemodynamically significant PDAs. Low magnitude systolic flow (SF) (*) suggests that there is no evidence for descending aortic obstruction.

**Table 1 children-07-00034-t001:** Comparative minimal ductal diameters data of various age groups.

Age in Months	TD MDD (mm)	Color-Doppler MDD (mm)	Angiographic MDD (mm) *
All patients: 4 to 303 months	3.99 ± 1.37	4.26 ± 1.57	2.28 ± 1.25
4 to 15 months	4.27 ± 1.36	5.23 ± 1.80	2.91 ± 1.50
16 to 30 months	3.49 ± 1.06	4.17 ± 1.61	1.89 ± 0.97
31 to 45 months	4.03 ± 2.04	4.07 ± 1.72	1.90 ± 1.19
46 to 60 months	3.76 ± 1.85	3.23 ± 1.22	1.46 ± 0.65
61 to 75 months	3.98 ± 1.36	3.83 ± 1.29	3.26 ± 1.69
76 to 303 months	4.12 ± 1.25	4.14 ± 1.34	2.05 ± 0.83

2D, Two-dimensional; MDD, Minimal ductal diameter. mm, millimeters; Mean ± Standard deviations are shown; * Angiographic MDDs were smaller (*p* < 0.05 to <0.01) than two-dimensional and color-Doppler echocardiographic MDDs for the entire cohort and for all six age groups. Reproduced from Rao P.S., et al. [29].

**Table 2 children-07-00034-t002:** Characterization size of Patent Ductus Arteriosus (PDA) in premature infants by echo-Doppler studies.

Parameter	Small PDA	Moderate PDA *	Large PDA *
Size of the LA	Normal	Mildly dilated	Moderate to severely dilated
LA:Ao Ratio	≤1.4:1	1.4 to 1.6	≥1.6
Size of the LV	Normal	Mildly dilated	Moderate to severely dilated
Systolic Function of the LV	Normal	Normal	Normal, hyper-contractile or diminished function
Estimated PA Pressure	Normal	Mildly elevated	Moderate to severely elevated
Minimal Diameter of the PDA	≤1.4 mm	1.4 to 2.0 mm	≥2.0 mm
Doppler Velocity across the PDA	High (3.0 to 4.0 m/s)	~2.0 m/s	Low (~1.0 m/s)
Descending Aortic Doppler Flow Velocity Pattern	Normal anterograde flow (Figure 33)	Normal anterograde flow (Figure 33)	Normal or absent anterograde flow or presence of retrograde flow (Figure 34)

* High probability for a hemodynamically significant patent ductus arteriosus (hsPDA) if associated with worsening of respiratory function or failure to wean from ventilatory support at an usual rate. Ao, aorta; LA, left atrium; LV, left ventricle; mm, millimeter; m/s, meters per second; PDA, patent ductus arteriosus; PA, pulmonary artery. Reproduced from Rao P.S., et al. [33].

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
