# Peer review of "The Author’s Contributions to Echocardiography Literature (Part II—1991–2020)"

_children, 2020, doi:10.3390/children7040034_

Round 1

Reviewer 1 Report

This review summarized the Author's Contributions to Echocardiography
Literature for all different topics which will help the readers to understand the various function of the echocardiography. This is a good review but I suggest to add a paragraph about the prospective future use or improvement of the echocardiograghy.

There are also minor changes need to make:

  1. Add the unit to tale 1.
  2. Change "shorting fraction" to "fraction shorting". Use FS as abbreviation
  3. Please describe how comparable are the LVFS and LVAS to measure the cardiac left ventricle contractile function.

Author Response

  1. The units in table 1, mm (millimeter) was added
  2. The reviewer suggested "Change "shorting fraction" to "fraction shorting". Use FS as abbreviation" - Done
  3. The reviewer suggested to describe how comparable are the LVFS and LVAS to measure the cardiac left ventricle contractile function. A short sentence was added to the text to address this suggestion

Reviewer 2 Report

Its a good review article with concise data. The author has contributed significantly to the field of pediatric cardiology and echocardiography. An interesting case of development of supravalvar pulmonary stenosis following the Nuss procedure.

Author Response

Thanks to reviewer. No revisions were suggested.

Reviewer 3 Report

This paper is a comprehensive review of the second half of the author's career and contribution to the literature.  Several of his studies contributing to the field of pediatric echocardiography are reviewed and nicely summarized.  Typically, at the end of each paper review, there is a short paragraph relating the present-day importance of the paper or a more current update;  how things have changed since the initial publication date.  Overall it is a nice review of an impressive career. 

I do not have significant changes to recommend.  The strengths of the paper include succinct and effective summaries of the published works as well as selected images for publication.  One small change I may add is in the review of the mixed TAPVR paper, the end paragraph should include the advantages of cross-sectional imaging in this diagnosis.

Author Response

Thanks to the reviewer for the complimentary remarks.

The reviewer suggested to add advantages of cross-sectional imaging in this diagnosis. I revised the manuscript accordingly.